# Preterm ETs Are Significantly Reduced Compared with Adults and Partially Reduced Compared with Term Infants

**DOI:** 10.3390/children9101522

**Published:** 2022-10-05

**Authors:** Aila Wirkner, Antje Vogelgesang, Ines Hegge, Anja Lange, Dirk Manfred Olbertz, Bernd Gerber, Matthias Heckmann, Johanna Ruhnau

**Affiliations:** 1Department of Neonatology and Pediatric Intensive Care, University Medicine Greifswald, 17475 Greifswald, Germany; 2Department of Neurology, University Medicine Greifswald, 17475 Greifswald, Germany; 3Department of Neonatology, South Clinic, University Rostock, 18059 Rostock, Germany; 4Department of Obstetrics and Gynecology, South Clinic, University Rostock, 18059 Rostock, Germany

**Keywords:** extracellular traps, preterm infant, neutrophils, monocytes

## Abstract

The release of DNA by cells during extracellular trap (ET) formation is a defense function of neutrophils and monocytes. Neutrophil ET (NET) formation in term infants is reduced compared to adults. Objective: The aim was to quantify NET and monocyte ET (MET) release and the respective key enzymes myeloperoxidase (MPO) and neutrophil elastase (NE) in preterm infants. In this prospective explorative study, ET induction was stimulated by N-formylmethionine-leucyl-phenylalanine (fMLP), phorbol 12-myristate 13-acetate (PMA), lipopolysaccharide (LPS), and lipoteichoic acid (LTA) in the cord blood of preterm infants (*n* = 55, 23–36 weeks) compared to term infants and adults. METs were quantified by microscopy, and NETs by microscopy and flow cytometry. We also determined the MPO levels within NETs and the intracellular concentrations of NE and MPO in neutrophils. The percentage of neutrophils releasing ET was significantly reduced for preterm infants compared to adults for all stimulants, and with a 68% further reduction for PMA compared to term infants (*p* = 0.0141). The NET area was not reduced except for when fMLP was administered. The amount of MPO in NET-producing cells was reduced in preterm infants compared to term infants. For preterm infants, but not term infants, the percentage of monocytes releasing ETs was significantly reduced compared to healthy adults for LTA and LPS stimulation. Conclusion: In preterm infants, ETs are measurable parts of the innate immune system, but are released in a reduced percentage of cells compared to adults.

## 1. Introduction

The development of the immune system during the first hours and days of life plays an important role in the development of an independent existence after birth, but the details are still not fully understood. Although the newborn is colonized with billions of microbes during birth, no overwhelming immune reaction is observed due to the balanced immune regulation seen in early life [1]. Nevertheless, preterm infants have a higher risk of developing infections [2]. In Europe, preterm birth rates range from 5.5 to 11.1% of all live births (from 4.3 to 8.7% for singleton births, and 42.2 to 77.8% for multiple births) [3]. In the U.S., a preterm birth rate of up to 10% is reported [4]. Despite improved medical neonatal intensive care management, preterm birth is still one of the main causes of perinatal morbidity and mortality [5]. Preterm infants have a higher susceptibility to infectious diseases, and early-onset infection especially contributes to morbidity and mortality in this cohort [6].

Granulocytes and monocytes play a central role in pathogen-associated molecular pattern recognition and the clearance of infections. However, preterm infants harbor an immature immune system, with reduced innate and adaptive immunity. Cord blood monocytes exhibit functional deficiencies, including a lower Toll-like-receptor (TLR) surface expression, impaired TLR signaling, and diminished cytokine production [7]. Furthermore, monocytes express less HLA-DR, which predicts sepsis development in neonates [8] and is accompanied by impaired phagocytosis [9]. In addition, neutrophils exhibit defects in migration and activation [10].

Extracellular trap formation is an additional defense mechanism. Upon bacterial stimuli, neutrophils extrude chromatin fibers loaded with antimicrobial proteins and enzymes, including myeloperoxidase (MPO) and neutrophil elastase (NE). These entities are called neutrophil extracellular traps (NETs) [11,12,13,14]. Microorganisms become entangled within these traps and are killed due to their proximity to high concentrations of antimicrobial proteins. The enzymes NE and MPO are stored in granules and act as important mediators to control infections after release [11]. NE and MPO colocalize in the extracellular DNA fibers of extracellular traps (ETs), which leads to the decondensation of intracellular chromatin and initiates ET formation [13]. In addition to the classical phagocytic function, monocytes and macrophages can release ETs that contribute to antimicrobial defense [15,16,17]. Nevertheless, in macrophages, this contribution is still poorly understood [18] as it was discovered later than in neutrophils and has been more rarely investigated.

As published previously in several studies, both the percentage and area of NET-forming cells are reduced in term infants in comparison to healthy adults [19,20,21]. In addition, the area covered by NETs is a measure of the effectiveness of the containment of bacterial spreading, which is poorer in term infants than in adults [19]. Whether preterm infants (<37 weeks) release even fewer and smaller NETs than term infants is unknown, as is the extent to which monocyte ETs (METs) are present in term and preterm infants.

Therefore, dependent upon gestational age (in term and preterm infants), this study analyzed NETs by microscopy and flow cytometry, and METs by microscopy only. The percentage of either total monocytes or neutrophils releasing NETs/METs is referred to as the percentage of NET/METs throughout the manuscript. Furthermore, we assessed the area covered by the released NET/METs from each individual cell. We hypothesized that ET formation in preterm infants is further reduced compared to term infants. A young healthy adult cohort served as a control. Furthermore, we quantified NE and MPO as key enzymes in NET formation.

## 2. Materials and Methods

### 2.1. Study Population

Preterm (<37 weeks) and healthy term infants (≥37 weeks) were recruited between July 2014 and October 2018 at the University of Medicine at Greifswald and the University of Medicine at Rostock in Germany. Gestational age was based on the date of the mother’s last menstruation and, if available, corrected using the first ultrasound measurement during pregnancy. Healthy adults (aged 18–33 years) served as controls. Exclusion criteria were severe congenital malformations, chromosomal aberrations, and a lack of written consent. Early-onset infection was defined as clinical symptoms and C-reactive protein > 10 mg/L within the first 72 h of life.

### 2.2. Collection of Umbilical Cord Blood

Immediately after birth, venous umbilical cord blood (mean blood volume of preterm = 2.0 mL) was filled into heparinized tubes (BD Vacutainer, Becton Dickinson GmbH, Heidelberg, Germany) by the midwives using standard blood collection sets by the same method used for peripheral blood collection. Blood was then stored at room temperature and processed within 2–3 h after birth.

For the measurement of intracellular MPO and NE amounts, blood was collected into additional EDTA tubes (BD Vacutainer, Becton Dickinson GmbH, Heidelberg, Germany).

### 2.3. NET Assay by Microscopy

Neutrophil granulocytes were prepared for microscopy as described previously [19]. Neutrophils were isolated from heparinized whole blood which was overlaid above 5 mL Histopaque^®^-1119 (Sigma Aldrich, St. Louis, MO, USA) and centrifugated for 20 min (2000 rpm). The pink neutrophil layer was removed and washed. The supernatant was discarded, and the cells were resuspended in Dulbecco’s phosphate-buffered saline (DPBS). To increase the purity of neutrophils, this cell suspension was overlaid above a Percoll gradient and centrifugated for 20 min (2000 rpm). The neutrophil layer was removed and washed. Cells were then resuspended in Hanks’ balanced salt solution (HBSS) and diluted to 5 × 10^4^ cells/mL. The cell suspension was plated into 24-well plates (1 mL/well) and allowed to rest for 30 min at 37 °C in a 5% CO_2_ atmosphere before adding the following different stimulators for NET induction for an additional 2 h: phorbol-12-myristate-13-acetate (PMA 0.736 µM, Glycotope, GmbH, Berlin, Germany), N-formylmethionine-leucyl-phenylalanine (fMLP; 0.4545 µM, Glycotope GmbH, Berlin, Germany), or lipopolysaccharide (LPS; 0.9 µg/mL, Sigma Aldrich, St. Louis, MO, USA). Unstimulated neutrophils were used as a negative control. After incubation, non-cell-permeant SYTOX Green (Thermo Fisher Scientific, Eugene, OR, USA) was added to detect extracellular DNA. APC Annexin V (Biolegend, San Diego, CA, USA) was used to stain apoptotic cells [19]. Peripheral blood from adults served as a control and was prepared accordingly. This isolation method usually yields unstimulated viable neutrophils with >90% purity. Experiments were performed under standardized conditions at an approximate pH of 7.5.

### 2.4. Microscopy Analysis of NETs

For each stimulated condition and unstimulated control, 10 fluorescent and phase-contrast images were evaluated on a LEICA DBMI-4000b microscope. To calculate the NET percentage, we calculated the ratio of the total number of all neutrophils and the ET-releasing neutrophils. The NET area was quantified as the fluorescent area covered by each cell using Fiji Software (Version 1.46) (GNU General Public License, Boston, MA, USA), which corresponds with the size of the NETs. NETs were defined as a fluorescent area ≥ 300 µm^2^ [19,22]. The fluorescence microscopy method enables NET analysis while excluding bias due to necrotic cells. Annexin V staining allowed us to clearly distinguish between NET formation, apoptosis, and necrosis. Overall (including all NET experiments), the percentage of apoptotic cells was comparable to data reported by Lipp et al. [19].

### 2.5. MET Assay

For the analysis of MET formation, umbilical cord blood was collected as described for ET staining. After PBMC isolation by standard Ficoll centrifugation, monocytes were negatively isolated using the Dynabeads^®^ Untouched™ Human Monocytes Kit (Invitrogen, Life Technologies, Carlsbad, CA, USA). Cells were isolated according to the manufacturer’s instructions. Monocytes were then resuspended in HBSS and diluted to 5 × 10^4^ cells/mL. The cell suspension was plated into 24-well plates (1 mL/well) and allowed to rest for 30 min at 37 °C in a 5% CO_2_ atmosphere. Afterwards, cells were stimulated by either PMA (50 nM; Tocris, Bio-Techne GmbH, Wiesbaden-Nordenstadt, Germany), lipoteichoic acid (LTA; 2 mg/mL, Sigma Aldrich, St. Louis, MO, USA), or LPS (2 mg/mL, Sigma Aldrich, St. Louis, MO, USA) for 13 h at 37 °C in a 5% CO_2_ atmosphere. For microscopy, cells were stained with non-cell-permeant SYTOX Green (Thermo Fisher Scientific, Eugene, OR, USA) for extracellular DNA, and the cell-permeant SYTO™ 60 red fluorescent nucleic acid stain (Thermo Fisher Scientific, Eugene, OR, USA). Syto60 was not used for the final evaluation of MET, solely as a dye within the established MET microscopy (shown in Appendix A).

The enrichment of monocytes was confirmed by identifying cells with the Hoechst 33342 Ready Flow™ Reagent DNA stain (Thermo Fisher Scientific, Eugene, OR, USA) and PE-Cy7 anti-human CD14 (Biolegend, San Diego, CA, USA) using flow cytometry. The enrichment of monocytes was always >84%. Experiments were performed under standardized conditions at an approximate pH of 7.5.

### 2.6. Microscopy Analysis of METs

The analysis of METs was performed by fluorescence microscopy as described above in Section 2.4, using Fiji Software (Version 1.46) (GNU General Public License, Bosten, MA, USA) METs were defined by a fluorescent area ≥200 µm^2^. In the literature, MET sizes are not as well-defined as NET sizes.

The analysis of METs by microscopy emerged as the most reliable method when using small amounts of blood in our research. We induced apoptosis and necrosis by H_2_O_2_ [23] to control for the mode of cell death, thereby defining a cut-off for MET sizes (shown in Appendix A). Afterwards, we detected two area peaks in our MET stimulation assays, as was shown for NETs. Therefore, we defined 200 µm^2^ to distinguish between the extracellular traps from necrotic or apoptotic monocytes.

### 2.7. NET FACS Preparation

NET quantification was also performed by flow cytometry according to a protocol introduced by Masuda et al. [24]. Neutrophils were isolated by a density gradient as described in Section 2.3, and 200 µL cell suspension was diluted in 800 µL PBS. Cells were stimulated as described above for NET detection. An unstimulated sample served as a control. Cells were stained with the Hoechst 33342 Ready Flow™ Reagent (Thermo Fisher Scientific, Eugene, OR, USA) and incubated in FACS tubes for 1.5 h at 37 °C in a 5% CO_2_ atmosphere. After incubation, cells were resuspended in 2 mL PBS and centrifugated for 5 min (1200 rpm). The supernatant was discarded, and each tube was filled with 200 µL of FACS buffer. Non-specific binding was blocked using the Human Fc Receptor Blocking Reagent, and cells were stained with anti-MPO-PE human antibody (Miltenyi Biotec GmbH, Bergisch Gladbach, Germany). A REA-control-PE human IgG1 antibody (Miltenyi Biotec GmbH, Bergisch Gladbach, Germany).) was used as the fluorescence minus one (FMO) control for MPO. The remaining erythrocytes were lysed with BD FACS Lysing Solution (BD Bioscience). Finally, SYTOX Green (Thermo Fisher Scientific, Eugene, OR, USA) was added to the samples. All tubes were placed on ice and measured immediately by a BD LSR II flow cytometer (BD Biosciences, Becton Dickinson GmbH, Heidelberg, Germany). FlowJo 10 was used to determine the percentage of NET formation by gating for the double-positive staining of MPO and SYTOX Green (shown in Appendix A).

Single-cell gating was performed by SSC-H/SSC-A and FSC-H/FSC-A to discriminate doublets. To detect all DNA-containing granulocytes, we used a SSC-A/Hoechst stain. Afterwards, we quantified NETs via the MPO and SYTOX Green-positive granulocytes (shown in Appendix A). No permeabilization steps were performed, since only extracellular MPO staining co-localizing with SYTOX Green identified NETs.

To quantify the amount of MPO per cell, we determined the mean fluorescence intensity (MFI) of MPO in quadrant 2 based on NET-defined events (shown in Appendix A). Experiments were performed under standardized conditions at an approximate pH of 7.5.

### 2.8. Intracellular NE and MPO Staining in Neutrophils

The amounts of intracellular NE and MPO were quantified using a BD LSR II flow cytometer (BD Biosciences, San Jose, CA, USA). Umbilical cord blood (EDTA) was collected immediately after birth (<2 h) as described before [25] and lysed, and cells were permeabilized in accordance with the instructions of the Human FoxP3 Buffer Set (BD Biosciences). Nonspecific binding was blocked by the Fc Receptor Blocking Reagent (Miltenyi Biotec GmbH, Bergisch Gladbach, Germany). MPO was stained by a FITC-labeled antibody by following the manufacturer’s instructions for the FITC anti-human flow kit (Biolegend, San Diego, CA, USA). For NE staining, cells were incubated with anti-human-NE as a primary antibody (DAKO, Glostrup, Denmark). PE-labeled goat anti-mouse immunoglobulin G (minimal-x-reactivity, Biolegend, San Diego, CA, USA) served as a secondary antibody. Monocytes were identified by APC anti-human CD14 antibody (Biolegend, San Diego, CA, USA). Flow cytometry data were analyzed by FlowJo 8.2.0. The amounts of NE and MPO per cell were quantified by MFI. Isotype and FMO controls were used to determine positive NE/MPO events.

### 2.9. Measurement of Arterial Cord Blood pH

Arterial pH was determined by an ABL 90 Flex Radiometer 090R0121N0001 (Radiometer GmbH, Krefeld, Germany) according to the manufacturer’s instructions immediately after birth. Here, arterial cord blood pH served as a measure for the clinical status of the newborn at birth rather than a variable to analyze the association of pH and NETs. In contrast, venous cord blood was used for the determination of NETs. Venous cord blood has a higher pH than arterial cord blood (mean 7.32 vs. 7.24) because it is oxygenated by the placenta. The pH in the venous cord blood was not determined.

### 2.10. Statistical Analysis

The sample size was calculated based on the effect sizes published by Lipp et al. [19]. To detect a NET reduction of 30%, a sample size of 10 infants in each group was calculated (α = 5%, ß = 10%, power 0.8).

Due to the fact that preterm infant recruitment was less frequent, the recruitment of term infants was performed continuously to avoid selection bias over time. The investigator was blinded to group allocation during data analyses.

All data sets were tested for adherence to Gaussian distribution using the Shapiro–Wilk test. Multiple comparisons of Gaussian-distributed data were performed using an analysis of variance and the Bonferroni correction for multiple comparisons as a post-test. As some of the data in each experiment failed the normality test, we used the non-parametric Kruskal–Wallis test with Dunn’s multiple comparisons as a post-test. Post-tests were only performed when initial testing revealed significant differences between groups. Correlations were determined by Spearman or Pearson analysis. GraphPad PRISM 8.2.0 (GraphPad Software Inc., San Diego, CA, USA) was used for all analyses. A *p*-value < 0.05 was considered significant. Results are presented as the median (minimum–maximum).

## 3. Results

### 3.1. Participant Characteristics

Table 1 presents the clinical characteristics and number of probands from whom blood was available for the different analyses. The numbers per group vary because the available material was not sufficient to include all the analyses for each sample. Detailed clinical characteristics of the preterm infants (*n* = 55, 23–36 weeks of gestational age) are given in Appendix A. Only one preterm infant of the NET FACS group had an early-onset infection, while the blood culture remained negative.

### 3.2. NETs and MPO within NETs in Preterm Infants in Comparison to Term Infants and Adults

In the microscopy assay, the percentage of NETs-releasing neutrophils was distinctly lower in term and preterm infants than in healthy adults for the unstimulated, LPS-, and fMLP-stimulated conditions.

A significantly reduced NET percentage was observed for preterm infants compared to term infants, with a 68.13% relative reduction for PMA (*p* = 0.0141; shown in Figure 1A). The NET area was not different between groups except for a reduced area in preterm infants compared to adults under fMLP stimulation (Figure 1B).

As determined by flow cytometry, term and preterm infants had a reduced percentage of NET formation after stimulation with PMA, fMLP, and LPS, as well as in unstimulated conditions compared to healthy adults. No difference was detected in the NET percentage between term and preterm infants (shown in Figure 2A).

Gestational age positively correlated with the percentage of NETs determined by microscopy after stimulation with PMA (shown in Figure 3C). The same tendency was observed for fMLP and LPS stimulation (shown in Figure 3E,G). NET area positively correlated with gestational age for fMLP-stimulated cells (shown in Figure 3F). Although PMA (shown in Figure 3D) also tended to be related to gestational age, unstimulated and LPS cells were not (shown in Figure 3A,B,H).

The amount of MPO within NETs, as determined by MFI, was lower in preterm infants compared to adults in unstimulated samples and fMLP and LPS-stimulated samples. Differences for term infants and adults were only detected in fMLP-stimulated neutrophils (shown in Figure 2B). Gestational age positively correlated with the amount of MPO within NETs for the unstimulated (r = 0.4201, *p* = 0.0291), PMA (r = 0.4002, *p* = 0.0386), and fMLP-treated samples (r = 0.5023, *p* = 0.0124).

Univariate analysis did not reveal any influence of the mode of delivery. The percentage of NETs tended to be higher in male infants than female infants when stimulated with LPS (*p* = 0.0632). With PMA and LPS stimulation, the NET area negatively correlated with the arterial umbilical cord pH as a measure of the clinical status of the newborn at birth (shown in Appendix A). No other correlations were found.

### 3.3. MPO/NE in Neutrophils

Our FACS NET results showed a reduced association of MPO and NETs in term infants in comparison to neutrophils of healthy adults. The intracellular amount of NE and MPO in neutrophils did not differ between term infants and healthy adults (shown in Appendix A). No other correlations were found for arterial umbilical cord blood pH.

### 3.4. METs in Preterm Infants in Comparison to Term Infants and Adults

The percentage of MET formation in term and preterm infants was reduced compared to healthy adults upon stimulation with PMA (shown in Figure 4A). For preterm infants, but not term infants, the percentage of METs was significantly reduced compared to healthy adults for LTA and LPS stimulation (shown in Figure 4A). Smaller METs were found only in preterm infants compared to adults in PMA-stimulated samples (shown in Figure 4B). No differences were found for MET area when comparing preterm and term infants.

For LPS stimulation, the percentage of METs correlated with gestational age (r = 0.4566, *p* = 0.0375).

We found no association between MET formation and the mode of delivery. However, a higher percentage of MET formation was measured in PMA-stimulated cells for male infants (*p* = 0.0053). A negative correlation was detected between arterial umbilical pH and the percentage of METs when stimulated with LTA or LPS (shown in Appendix A).

## 4. Discussion

The production of ETs is an important antimicrobial mechanism of the innate immune system. Accordingly, the failure to form NETs is associated with a severe form of immunodeficiency [26]. We have previously described a reduction in the percentage of NET-releasing neutrophils in term infants, as well as a lower percentage of NET-releasing neutrophils in a small group of late preterm infants compared to healthy adults [19]. Our current study extends these findings to MET formation and a larger group of preterm infants (*n* = 55) comprising the complete range of gestational age from 23 to 36 weeks. The strengths of this study also include precisely timed blood collection. Moreover, we studied the effects of different NET and MET formation stimuli (PMA, LPS, LTA, and fMLP) and observed a wide range of potentially relevant clinical factors.

The percentage of NET-releasing neutrophils was distinctly lower in term and preterm infants than in healthy adults when determined by microscopy in unstimulated conditions and under different stimuli. As hypothesized, the percentage of NETs released by preterm infants was significantly lower in comparison to term infants for PMA, but this finding could not be extended to other stimuli. Only fMLP stimulation induced a lower NET area released by preterm infants in comparison to adults. Furthermore, the correlation of gestational age showed an influence of PMA on the NET percentage. Using flow cytometry as a second NET-detecting method, these findings were confirmed by a higher percentage of NETs and a higher amount of MPO in NETs from adults compared to preterm infants. Differences between term and preterm infants were not observed. Although this method was already published as a promising approach [24], it has some limitations; released ETs are not fixed to the cell membrane, which makes it more difficult to detect them by flow cytometry [24,27].

The data suggest that even neutrophils of preterm infants can produce NETs but at a considerably reduced percentage compared to adults. These results may be explained by the different NET and MET formation stimuli (PMA, LPS, LTA, and fMLP) [16,20,21,28]. We used fMLP, which mimics the N-formyl oligopeptides released by bacteria and activates circulating blood leukocytes by binding to specific G protein-coupled receptors on these cells. Isolated LTA (derived from Gram-positive bacteria) and LPS (derived from Gram-negative bacteria), both bacterial-wall proteins, have been described to induce NETs [14]. LPS simulates pathological stimulation by signaling via Toll-like receptors (TLRs) 2/4. Our results indicate only a tendency towards a difference in NET reduction between term and preterm infants with these more natural stimuli (LPS, LTA, and fMLP). Hoppenbrouwers et al. even found no induction of NETS by LPS, whereas PMA was a consistent and potent inducer of NETs in vitro [29].

However, PMA is not physiologically relevant, since it does not activate physiological processes in vivo [29]. Therefore, our study has some limitations. Differences between term and preterm infants were mainly related to stimulation by PMA in our ex vivo experiments. Stimuli such as LPS and fMLP are also expected to trigger an immune response in vivo, which could trigger alternate pathways that induce NETs. Furthermore, we did not study living bacteria as a strong natural stimulus of ETs compared to PMA [29]. Nonetheless, to our knowledge, there is only one comparable study with respect to the number (*n* = 39) of preterm infants [30]. In contrast to our study, NETs were studied in tracheal aspirates to correlate with bronchopulmonary dysplasia.

Campbell et al. show a higher level of high-temperature requirement serine protease A1 (HTRA1) in placental blood as a possible explanation of our findings published in 2017, in which we showed reduced NET production in term infants [19,31]. Furthermore, neonatal NET-inhibitory factor (nNIF) and NIF-related peptides (NRPs) could influence the altered NET-production after birth as shown by Yost et al. [32]. A1ATm358, another placental-related modulatory factor found in the perinatal milieu, is no longer active in neonates after birth [32]. Only recently has it been shown that the group of NET inhibitors improves survival in experimental neonatal infectious peritonitis [33].

Another reason for the diminished percentage of MET formation in preterm infants might be the altered reactive oxygen species (ROS) production in term infants as one key element of NET/MET formation [34]. In one of our studies, we found that neutrophils from term infants have partly diminished ROS production, especially in pro-inflammatory granulocytes, although the amount of ROS per cell is partly enhanced by *Escherichia coli* or fMLP stimulation [25]. Whether the diminished NET/MET formation is mediated by ROS production and/or a more proximal mechanism of ROS induction should be analyzed in future studies.

Further studies have revealed differences between the blood of preterm and term infants: Anderson et al. reported reduced frequencies of monocytes and CD56^bright^ NK cells, as well as CD8+ T-cells, and γδ T-cells in preterm infants [35]. Moreover, controversial data exist regarding the amount of proinflammatory interleukin-6 (IL-6) in cord blood: While Anderson et al. reported elevated levels in preterm infants, Sullivan et al. detected lower levels in comparison to term infants [36]. Whether these balanced immune alterations increase the susceptibility of infections in preterm infants should be an object of future studies.

While a reduction in NETs/METs may influence bacterial killing, we only investigated cells ex vivo. Our findings might therefore not necessarily reflect the in vivo situation. Due to the low number of neonatal early-onset infections (C-reactive protein > 10 mg/L and symptoms of an infection during the first 72 h of life) in our observed cohort, we were not able to analyze the direct effects of NETs/METs on neonatal infections. Webster et al. [37] showed an increase in METs following exposure to *Escherichia coli* or *Klebsiella pneumoniae* ex vivo, which was dependent on the bacterial load. NET formation and nuclease activity are decreased in septic patients [38]. Therefore, our findings indicate that neonates may have a lower capacity to fight these bacteria. In contrast, Stiel et al. [39] reported that NET markers (e.g., NE, MPO, and cell-free DNA) in umbilical cord blood do not appear to be a predictor of the onset of neonatal sepsis within 72 h postpartum. This is in contrast with newer experimental data from Colón et al., which showed that infant C57BL/6 mice subjected to sepsis or LPS-induced endotoxemia produced significantly higher levels of NETs than adult mice [40]. Nevertheless, these data lack proof of biologically active NETs in a cohort of neonatal infections and are in line with our neutrophil levels of MPO/NE from neonates, which were similar to the levels in adults. Levy et al. demonstrated an equal level of MPO in term neonates and adults [41]. In regard to this point, term infants seem to have some of the same requirements for decondensed chromatin fibers as healthy adults.

Limitations: A weakness of this study is the lack of intracellular staining of citrullinated histone H3, which is a known part of ETs. Cord blood volume is very limited, especially in the placentas of preterm neonates. In preterm infants, our microscopy assay alone demands a minimum of 500 µL cord blood. Due to the different composition of adult and preterm leukocytes, as well as the immature complement system in preterm probands, the isolation of neutrophils and detection of extracellular traps via methods developed for adult samples was challenging. Therefore, we prioritized the microscopy approach, where we could produce consistent data with very low amounts of cord blood. Additionally, where sufficient cord blood could be obtained, flow cytometry could be performed to confirm the microscopy results. METs were only detected by fluorescence microscopy because the release of MPO into the extracellular space could not, unlike neutrophils, be detected in monocytes in our research. Data from Depreester et al. suggest the high biological variability in individuals and the higher expression of MPO in PMN than in monocytes as possible causes [42].

## 5. Conclusions

In preterm infants, NETs and METs are parts of the innate immune system, but are considerably reduced in percentage compared to adults. Compared to term infants, decreasing gestational age is related to a further reduction in ET formation with the non-physiological stimulus PMA. Only fMLP induced a difference in NET area in comparison to preterm and term infants, whereas induction with other stimuli led to similar NETs/METs areas and contents of MPO in both term and preterm infants. Further studies should investigate the importance of NETs/METs formation in clinical settings of bacterial infection.

## Figures and Tables

**Figure 1 children-09-01522-f001:**
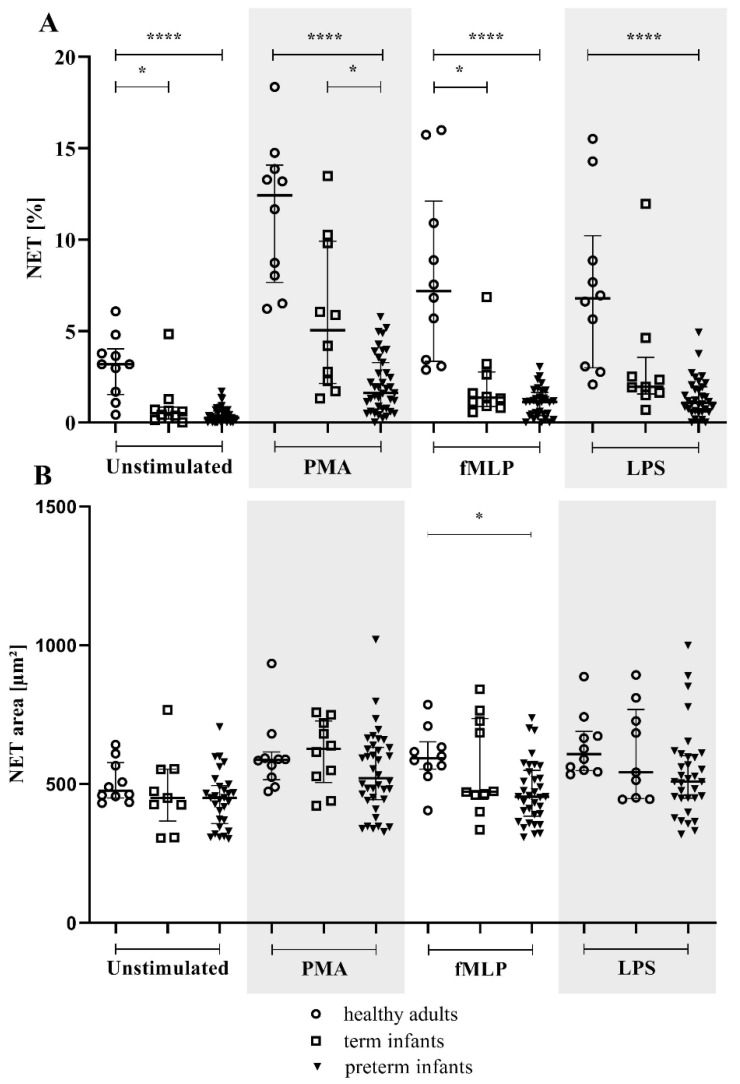
NET percentage and area in the preterm, term, and healthy adult groups. The NET-forming neutrophils of term (*n* = 10; light squares) and preterm (*n* = 39; light triangles) infants are shown compared to healthy adult subjects (*n* = 10; light dots). Isolated neutrophils were either unstimulated or treated with phorbol 12-myristate 13-acetate (PMA), N-formylmethionine-leucyl-phenylalanine (fMLP), or lipopolysaccharide (LPS). (**A**) NET percentage and (**B**) NET area (µm^2^) were measured by microscopy (NET >300 µm^2^). * *p* < 0.05, **** *p* < 0.0001. Data are presented as the median and interquartile ranges.

**Figure 2 children-09-01522-f002:**
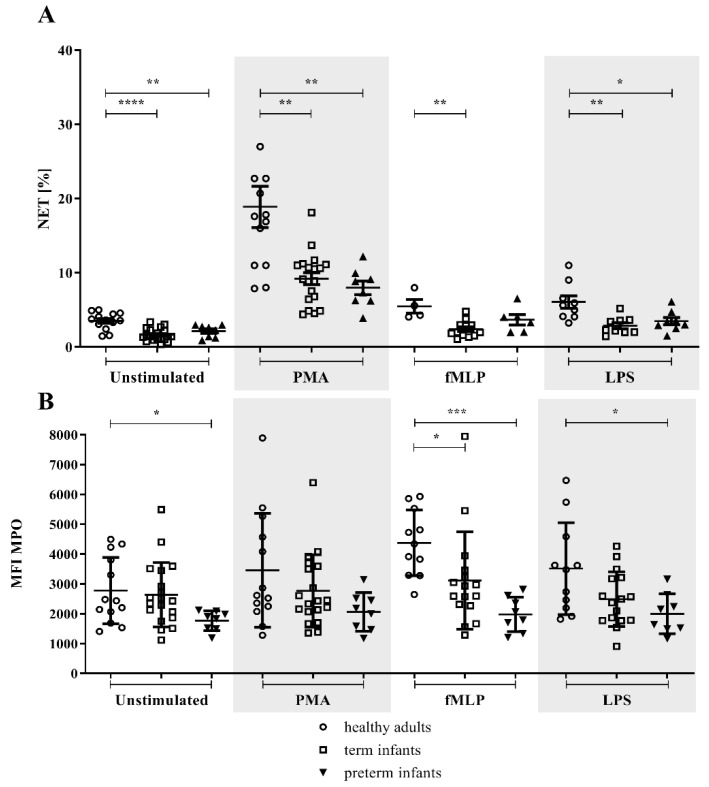
NETs were measured by flow cytometry in the preterm, term, and healthy adult groups. The NET-forming neutrophils of term (*n* = 19; light squares) and preterm (*n* = 8; light triangles) infants are shown compared to healthy adult subjects (*n* = 13; light dots). Isolated neutrophils were either unstimulated or treated with phorbol 12-myristate 13-acetate (PMA), N-formylmethionine-leucyl-phenylalanine (fMLP), or lipopolysaccharide (LPS). (**A**) The NET percentage was measured by flow cytometry. (**B**) The MPO amount in NETs was determined by the mean fluorescence intensity (MFI). * *p* < 0.05, ** *p* < 0.01, *** *p* < 0.001, **** *p* < 0.0001. Data are presented as the median and interquartile ranges.

**Figure 3 children-09-01522-f003:**
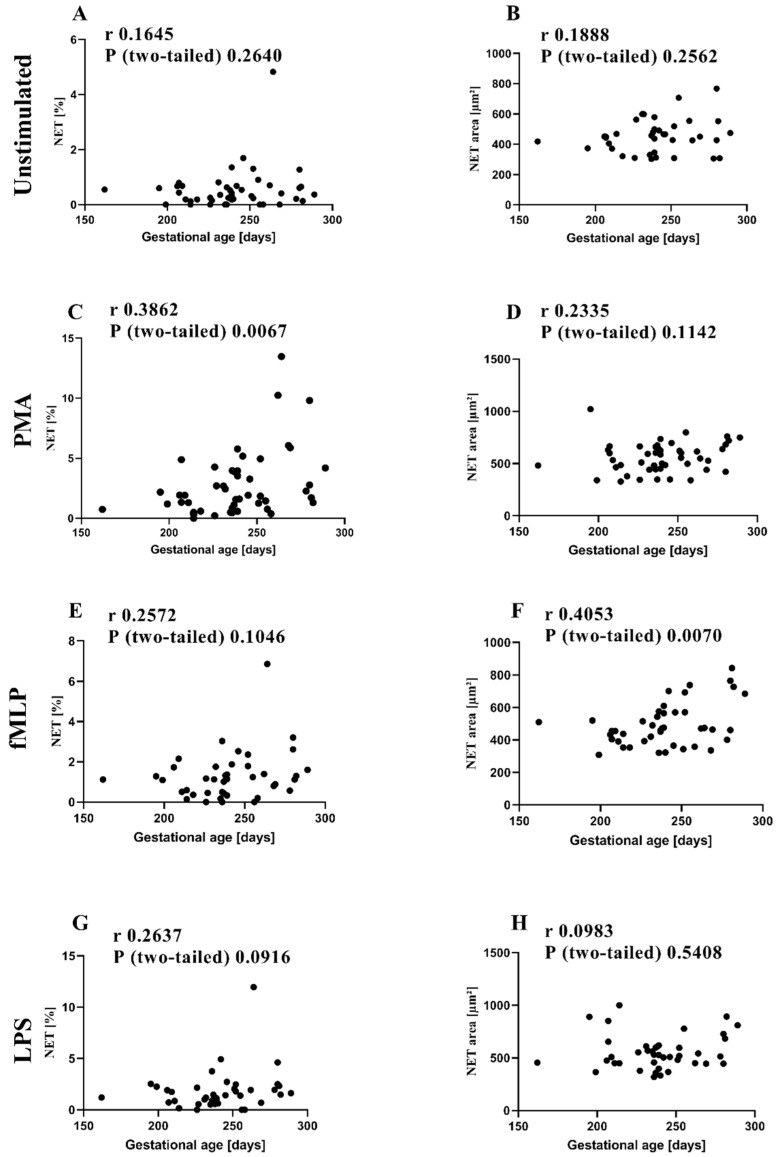
Correlation of NET percentage and area with gestational age. Preterm (*n* = 39) and term (*n* = 10) infants were investigated for a correlation by Spearman analysis. (**A**,**B**) Isolated neutrophils were either left unstimulated or (**C**,**D**) treated with phorbol 12-myristate 13-acetate (PMA), (**E**,**F**) N-formylmethionine-leucyl-phenylalanine (fMLP), or (**G**,**H**) lipopolysaccharide (LPS).

**Figure 4 children-09-01522-f004:**
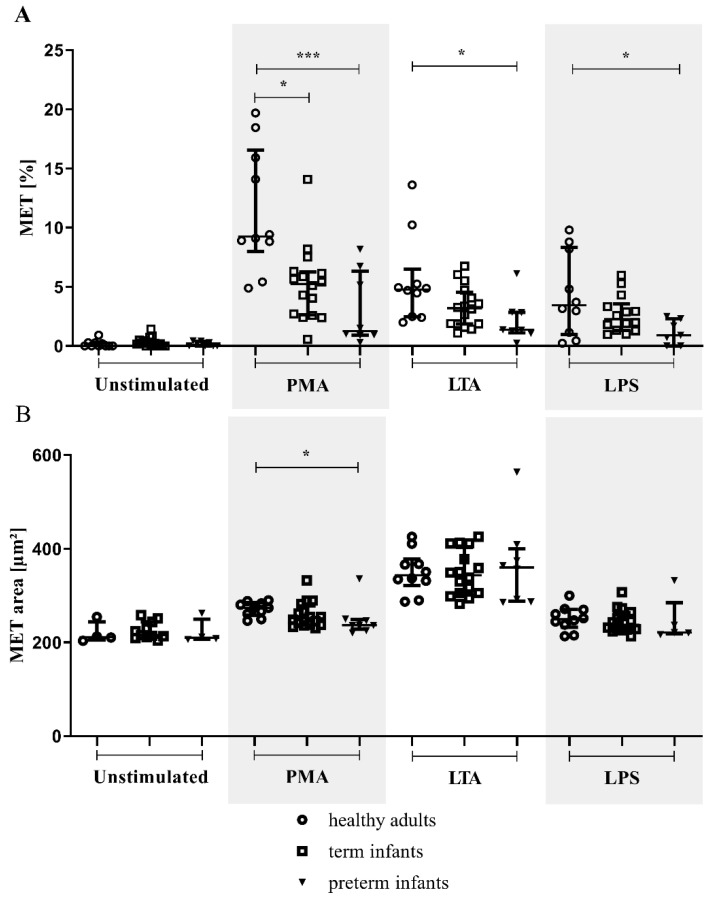
MET percentage and area in the preterm, term, and healthy adult groups. The MET-forming monocytes of all term (*n* = 16; light squares) and preterm (*n* = 8; light triangles) infants are shown compared to healthy adult subjects (*n* = 10; light dots). Isolated monocytes were either unstimulated or treated with phorbol 12-myristate 13-acetate (PMA), lipoteichoic acid (LTA), or lipopolysaccharide (LPS). (**A**) MET percentage and (**B**) MET area (µm^2^) were measured by microscopy; (MET >200 µm^2^). * *p* < 0.05, *** *p* < 0.001. Data are presented as the median and interquartile ranges.

**Table 1 children-09-01522-t001:** Characteristics of analyzed populations for NETs and METs assays. Gestational age, birth weight, arterial umbilical pH, and APGAR are illustrated as the median (min–max).

	Population	N	Female	Gestational Age, Weeks	Birth Weight, g	Arterial Umbilical pH	APGAR 5-min	Cesarean Section	Spontaneous Delivery
NET microscopy	Preterm	39	20 (39)	33 + 5 (23 + 1–36 + 3)	1950 (600–3150)	7.34 (7.14–7.45)	9 (6–10)	26 (39)	13 (39)
	Term	10	5 (10)	39 + 5 (37 + 3–40 + 2)	3190 (2500–4970)	7.28 (7.13–7.34)	10 (9–10)	6 (10)	4 (10)
	Adults	10	8 (10)	n.a. *	n.a.	n.a.	n.a.	n.a.	n.a.
NET FACS	Preterm	8	3 (8)	33 + 4 (31 + 0–35 + 2)	2262.5 (1480–2450)	7.33 (7.29–7.36)	9 (8–10)	5 (8)	3 (8)
	Term	19	10 (19)	40 + 0 (39 + 0–41 + 5)	3680 (2640–4485)	7.28 (7.13–7.36)	10 (8–10)	11(19)	8 (19)
	Adults	13	8 (13)	n.a.	n.a.	n.a.	n.a.	n.a.	n.a.
MET microscopy	Preterm	8	2 (8)	32 + 4 (26 + 2–36 + 1)	1917.5 (990–2950)	7.33 (7.28–7.42)	9 (8–10)	6 (8)	2 (8)
	Term	16	10 (16)	40 + 0 (37 + 4–41 + 5)	3495 (2640–4830)	7.31 (7.14–7.36)	10 (8–10)	10 (16)	6 (16)
	Adults	10	5 (5)	n.a.	n.a.	n.a.	n.a.	n.a.	n.a.
NE/MPO FACS	Term	10	5 (10)	40 + 0 (40 + 0–40 + 0)	3483 (2720–3915)	7.32 (7.23–7.38)	9.5 (8–10)	7 (10)	3 (10)
	Adults	10	6 (10)	n.a.	n.a.	n.a.	n.a.	n.a.	n.a.

* n.a., not applicable.

## Data Availability

The datasets generated and/or analyzed during the current study are not publicly available due to data protection but are available from the corresponding author upon reasonable request.

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
