# Peer review of "Preterm ETs Are Significantly Reduced Compared with Adults and Partially Reduced Compared with Term Infants"

_children, 2022, doi:10.3390/children9101522_

Round 1

Reviewer 1 Report

Review of neutrophil and monocyte extracellular traps in preterm infants by Wirkner et al.

Major comments;

-          In this study NETs are quantified by microscopy and flowcytometry, however there is no information given if these methods correlate and give the same results. Moreover, what is the best method?

-          It is unclear how NET area is determined, especially in the unstimulated samples. Does this reflect the amount of total neutrophils? Also if NET area is not different between the groups it is not clear how fMLP stimulated NET area can correlate with gestational age.

-          In the discussion lines 336-338 seem out of place here.

-          The authors mention their attempts to measure citH3 and a NET elisa, but do not give details what went wrong. Is there no citH3 in cord blood cells?

-          Figure 1: what do the pictures of plain film add?

-          Why was the intracellular MPO and NE staining performed? What did the authors expect?

-          Many of the references are not very recent, maybe the authors should update this.

Minor comments;

Line 51: Neutrophil not with a capital N

Line 89: 10^4 instead of 104

Lines 92 and 93: explain the unusual concentrations for PMA and fMLP, and comment on other concentration used for METs

Figure 3A: IS there a difference in LPS stimulated cells between healthy adults and term infants?

Line 385: decreasing gestational age is related to …. , instead of contributes to….

Line 386: the unphysiological stimulus PMA, whereas…

Line 386: until area is further explained, it should be left out of the conclusion.

Reviewer 2 Report

This manuscript evaluates the ability of preterm infants to produce neutrophil extracellular traps (NETs) and monocyte extracellular traps (METs), compared to term infants and adults. ETs were induced with fMLP, PMA, LPS, and LTA in cord blood of preterm and term infants, and compared to adult peripheral blood. NET number was reduced for preterm infants compared to adults, and in preterm infants compared to term when stimulated with PMA. Average NET size was only reduced in preterm infants when simulated with fMLP. MET release following LPS and LTA stimulation was reduced in preterm infants compared to term and adult.

 Major:

 1.       In the abstract, authors mention METs are evaluated by microscopy, and NETs by microscopy and flow cytometry. However, Line 115 describes flow cytometry for MET analysis. Please revise.

 2.       Why were neutrophils stimulated for only 2 hours, and monocytes for 13?

 3.       The methods are confusing as written. Sections 2.2 and 2.3 appear as if authors are isolating neutrophils and monocytes, and then performing staining. However, 2.4 sounds like the cell types are combined and the classification of neutrophils and monocytes is only by size of the ET? If so, how can authors be sure what they think are METs are not just smaller NETs due to prematurity, etc.? 

Minor:

 1.       Lines 89 and 104, cell number needs to have the 4 as an exponent.

 2.       What is the specific Fiji software or extension used to analyze the ET size in Line 128?

Reviewer 3 Report

Review  1890751

Manuscript entitled "Neutrophil and monocyte extracellular traps in preterm infants" by Wirkner and Ruhnau. Study Neutrophils and Monocyte Extracellular Traps (NET and MET) of cells derived from preterm infants. Cells were collected from the umbilical cord of preterm or term infants and adults. Stimulation in vitro revealed that preterms have reduced effector function by Neutrophils or Monocytes.

It is well written, good paper, bringing original human data that is of interest. It can be accepted pretty much as is, minor comments are below.

Strength- primary human data.

Weakness- cord blood is known to have less cellular-immune activities than adults.

Minor Comments: please consider these point, do not hesitate to change only part. The paper is good, you may reject/explain any comment.

1.       Methodology may better describe the isolation of Neutrophil, this is a critical and sensitive step – what author refer as "standard" may vary among laboratories.

2.       Line 89 "… diluted to 5 x 104…" miss superscript 4.

3.       Figure's caption may better have titles that say the main finding, not the technique used. Authors may consider helping the reader to get their findings easy (eg- figure#3 may title: "Preterm NETs are significantly reduced compared with adults, and partially reduced compared with term infants.")

4.       Representative FACS plots are awesome! It might be better to move figures 1,2 to supplementary, or even better – include as panels with the quantified data.

5.       The discussion keeps limited to Neutrophils and Monocyte, authors may discuss other immune cells (eg NK) that were published to have reduced effector activities in the cord blood of infants.

Reviewer 4 Report

In this study, Wirkner A et al. investigate the capacity of umbilical cord blood derived from pre-term and term infants to undergo NET and MET formation, comparing these findings to a cohort of young healthy adults. The authors observe that NET and MET formation is less accentuated in both infant groups compared to adults in response to various stimuli (especially PMA), with some ex vivo experiments indicating decreased ET release in pre-term infants compared to infants delivered at term. While the study sheds light on pre-term infants, thus being novel with regards to NETosis and METosis, the manuscript has several substantial limitations that preclude publication.

1)     There should be a specific subsection of the Methodology where processing of umbilical cord blood is described. How was blood harvested from the cords? How much on average? How was it centrifuged? How was it stored? How was it anti-coagulated (it seems that for some experiments, EDTA was used, and for some, heparin)?

2)     Looking up reference 19 to find out how exactly neutrophils were isolated yielded only this: “Neutrophils were isolated from heparinized whole blood using standard Histopaque/Percoll gradient centrifugation”. This is not an adequately described methodology. Please give details, or provide another citation.

3)     Why were METs not investigated using flow cytometry, too?

4)     Early onset infection was defined as clinical symptoms and CRP > 10 mg/L in this trial. Please specify symptoms of infection; also, is CRP elevation exclusively confined to infection after birth? Since the immune system of newborns is, all of a sudden, confronted with pathogens yet unencountered, would there not be a general upregulation of the immune system, i.e., CRP elevation, as well? Did the authors consider measuring other markers of inflammation, e.g. procalcitonin? CRP is a ”slow” parameter, often rising long after onset of infection. IL-6 is given in a supplementary table yet never discussed.

5)     NET FC: neither Sytox Green nor MPO are specific for NET formation; why did the authors not employ a specific marker like citrullinated histone H3? Also, why were doublets excluded? Since NET formation leads to drastic changes in cell morphology / integrity and induces adhesion of neutrophils onto each other due to the DNA filaments, doublets should not be excluded.

6)     Regarding citH3, the authors acknowledge themselves that not having stained for citH3 is substantial limitation of this study. They cite low blood amounts harvested from cords as a major reason, yet then state that with microscopy and flow cytometry, “consistent data with low amounts of blood” could be produced. It seems to me that this would entail staining citH3, as well.

7)     If acidosis influences NET formation (as indicated in Fig. S3), how can the authors exclude that the differences observed between healthy adults and infants are not primarily driven by pH? Judging by the correlation plots, at least 75% of infants appear to be acidotic / anaerobic. Also, the authors measured pH with an ABL 90 Flex Radiometer. Did this device also measure lactic acid?

8)     Does umbilical cord blood resemble the circulation of the infant itself?

9)     In Table S1, the authors give information on the rate of respiratory distress syndrome after birth. Did they look for any connection to NETosis?

10)  In the discussion, the authors state that “even neutrophils of preterm infants are able to produce effective NETs”; yet functional assays are not performed. Therefore, effectiveness of released NETs can not be inferred.

11)  Line 115: The authors state that “enrichment of monocytes was always >75%” for MET assays. I find this purity to be surprisingly low. Did the authors investigate which cells are constituting the remaining 25%? Could neutrophil contamination bias the observed extent of METosis?

Figures

Figure 1: If the authors indeed induced METs, then why are there no filaments observed? The authors only show Sytox Green-positive cells. Since Sytox Green is cell-impermeant, the authors might have just demonstrated that in response to PMA, cells died. In this regard, may the PMA stimulation have shown that monocytes of term and pre-term infants are more resistant to PMA?

Figure 2: It seems that in adults, there are two separate populations, the second of which (more positive for Sytox) is not present in infants. What are these cells?

Figure 3: statistical testing for the response of neutrophils to stimuli are not provided here (e.g. unstimulated vs. PMA). Judging from the graph, there appears to be difference, of course; however, please clarify whether statistical testing was performed in a paired manner, too. NETs in percent (A) and NET area (mm2) are intriguingly discrepant. Why would the positivity of NETs be so different, but not the area? It appears that incubation time was too short: after 2 h, neutrophils initiate cell death, but expulsion of NETs has not occurred yet.

Figure 5: remove lines from correlation plots, as this implies regression. Formatting seems inconsistent (font size, axes thickness).

Minor points

Line 42: do the authors mean “pathogen-associated molecular pattern” (PAMP) recognition? If so, they should add the “molecular” since PAMP is an easily recognizable and well-known concept.

Line 62: clarify what percentage and area refer to, or choose a more general wording like “capacity to undergo NETosis”.

Table 1: why is “female” in adults not applicable?

Fig. S2: please also show negative / FMO stainings, so that gating thresholds are more easily apparent.

Fig. 4: please check whether data are in fact given as median [IQR]. To me, it looks like the mean is plotted?

Line 337: nNIF and NRPs have not been introduced so far.

Round 2

Reviewer 1 Report

The issues raised by the reviewer have been adequately answered.

Author Response

The paper was again revised for spelling and grammar

Reviewer 4 Report

In my opinion, the manuscript has undergone some major improvements since the last revision, and I thank the authors for addressing reviewer questions in detail.

Some questions / comments remain:

Regarding comment #2:

Please clarify whether Percoll was also used for isolating neutrophils (as indicated in V1 of the manuscript). Also see line 126: "After standard Histopaque/Percoll gradient centrifugation...".

Regarding comment #3:

I fail to understand why the authors cite two studies that predate the discovery of NETs (1987 and 1996) to explain why METs have a lower amount of co-localized MPO in extracellular traps. In these two studies, MPO content was measured three or five days after initiation of culture to demonstrate loss of MPO. The authors performed experiments within hours of blood draw. The reasoning for not measuring METs by flow cytometry is not provided in the revised version of the manuscript.

Regarding comment #5:

Please clarify why a HOECHST-positive cell is automatically still alive. As I understand, HOECHST does not assess cell viability…? According to a short internet search, HOECHST is membrane permeable. As such, doublets should be positive, not negative?

Also, if you used HOECHST positivity for gating cells that are still alive, and performed no permeabilization, cells gated afterwards cannot be Sytox-positive since Sytox explicitly does not permeate cell membrane. Thus, please clarify that there was no fixation or permeabilization step performed whatsoever.

Washing steps are not indicated in the methodology. Also, centrifugation settings are not provided (in our own lab, we centrifuge for 15 min at 3200 g, thus suffering only little losses after each washing step), as are the amount of blood used for each staining (for our group, it's 150 µl per sample). Similarly, the average yield of cord blood is not specified in the revised version of the manuscript.

Regarding comment #7:

Did the authors measure pH value of healthy adults blood? If so, correlating it with ex vivo NET formation could shed more light onto a possible pH-mediated effect on the observed differences. Also, was umbilical cord blood from infants venous or arterial?

Regarding Figure 1 / MET assay methodology

The authors provide stringent arguments for using non-strand-ETs for the MET assay ("Therefore, we defined 200 µm2 to distinguish extracellular traps from necrotic or apoptotic cells specific for monocytes" etc.). Yet, this should also be added to the manuscript, not only to the response letter, as this is viable information.

Round 3

Reviewer 4 Report

I thank the authors for their thorough revision. I have no further comments.